# Exopolysaccharides of *Bacillus amyloliquefaciens* Amy-1 Mitigate Inflammation by Inhibiting ERK1/2 and NF-κB Pathways and Activating p38/Nrf2 Pathway

**DOI:** 10.3390/ijms231810237

**Published:** 2022-09-06

**Authors:** Wei-Wen Sung, Yun-Yu Lin, Shen-Da Huang, Hsueh-Ling Cheng

**Affiliations:** Department of Biological Science and Technology, National Pingtung University of Science and Technology, Pingtung 912301, Taiwan

**Keywords:** anti-inflammation, anti-oxidation, *Bacillus amyloliquefaciens*, exopolysaccharides, immune modulation, MAPK, Nrf2, p38

## Abstract

*Bacillus amyloliquefaciens* is a probiotic for animals. Evidence suggests that diets supplemented with *B. amyloliquefaciens* can reduce inflammation; however, the underlying mechanism is unclear and requires further exploration. The exopolysaccharides of *B. amyloliquefaciens* amy-1 displayed hypoglycemic activity previously, suggesting that they are bioactive molecules. In addition, they counteracted the effect of lipopolysaccharide (LPS) on inducing cellular insulin resistance in exploratory tests. Therefore, this study aimed to explore the anti-inflammatory effect and molecular mechanisms of the exopolysaccharide preparation of amy-1 (EPS). Consequently, EPS reduced the expression of proinflammatory factors, the phagocytic activity and oxidative stress of LPS-stimulated THP-1 cells. In animal tests, EPS effectively ameliorated ear inflammation of mice. These data suggested that EPS possess anti-inflammatory activity. A mechanism study revealed that EPS inhibited the nuclear factor-κB pathway, activated the mitogen-activated protein kinase (MAPK) p38, and prohibited the extracellular signal-regulated kinase 1/2, but had no effect on the c-Jun-N-terminal kinase 2 (JNK). EPS also activated the anti-oxidative nuclear factor erythroid 2–related factor 2 (Nrf2) pathway. Evidence suggested that p38, but not JNK, was involved in activating the Nrf2 pathway. Together, these mechanisms reduced the severity of inflammation. These findings support the proposal that exopolysaccharides may play important roles in the anti-inflammatory functions of probiotics.

## 1. Introduction

Inflammation involves a series of defensive reactions, including removal of impaired tissues, and fighting against invading pathogens or harmful factors. However, prolonged or excessive inflammation results in tissue damage and prompts the development of several chronic disorders, such as cancers, type 2 diabetes, and cardiovascular diseases [1]. Probiotics have been suggested to modulate immune reactions and ameliorate inflammatory disorders, such as inflammatory bowel diseases [2,3]. Nevertheless, the mechanisms underlying these effects are not fully understood.

*Bacillus amyloliquefaciens*, a Gram-positive bacterium, has been suggested to be probiotic and prebiotic for animals [4,5,6,7,8], and GRAS (generally recognized as safe) for humans [9]. It is ubiquitously present in food and the environment, often used in food fermentation [8,10], and is present in human gut microbiota [11]. Recently, the evidence suggests that diets supplemented with *B. amyloliquefaciens* can reduce inflammation. For example, dietary supplementation with *B. amyloliquefaciens* improved the growth performance and decreased intestinal inflammatory responses in piglets with intra-uterine growth retardation [12]. Likewise, broilers fed with a diet supplemented with *B. amyloliquefaciens* SC06 showed decreased levels of the pro-inflammatory cytokines interleukine-6 (IL-6) and tumor necrosis factor-α (TNF-α) in the ileum [13]. Moreover, *B. amyloliquefaciens* administration reduced biomarkers of inflammation in dextran sulfate sodium-induced colitis in mice [14]. Although it was suggested in these studies that modulation of the intestinal microbiota was a potential mechanism underlying the anti-inflammatory effect of *B. amyloliquefaciens*, the exact molecular mechanism is still unclear. Nonetheless, a potential clue is that the extracellular extracts of *B. amyloliquefaciens* MBMS5 were reported to possess antioxidant activity and anti-inflammatory property [15]. Thus, the metabolites or secreted molecules of *B. amyloliquefaciens* may play a vital role in the anti-inflammatory function of this probiotic. In line with this, scholars have proposed that exopolysaccharides produced by probiotics likely have immunomodulatory and/or anti-inflammatory functions [16,17].

Microbial exopolysaccharides are carbohydrates synthesized and accumulated extracellularly by micro-organisms [17,18,19]. Structures of the exopolysaccharides produced by *B. amyloliquefaciens* amy-1 have been characterized in our previous study [20]. They are composed of mannose, glucose, and galactose in a ratio of 5:2:1 and are highly branched, whereas the linkage modes between the constituent monosaccharides have not been resolved. The molecular weights of most of the polymers are larger than 1000 kDa, with a major component estimated to be 1926.6 kDa. Furthermore, the exopolysaccharide preparation of amy-1 (EPS) was found to have hypoglycemic activity [20]. In our exploratory experiments, we tested the effect of EPS on the insulin resistance of cells induced through the proinflammatory lipopolysaccharide (LPS) of *Escherichia coli*, because the occurrence of type 2 diabetes is closely associated with chronic inflammation-induced insulin resistance [9,21]. Consequently, EPS ameliorated cellular insulin resistance (unpublished data), indicating that EPS may counteract the effect of LPS. We also found that EPS activated bitter taste receptors [22], whereas these receptors have been proposed to be involved in immune modulation [23,24,25]. Based on these observations, we hypothesize that amy-1 exopolysaccharides have anti-inflammatory activity. In this study, this hypothesis was tested, and the underlying mechanisms were explored.

## 2. Results

### 2.1. Anti-Inflammatory Effect of EPS

Exopolysaccarides were isolated from the medium of amy-1 culture. The total carbohydrates in this preparation (i.e., EPS) occupied 95.65 ± 0.68% of the total mass, while proteins, lipids, and polyphenols were undetectable in EPS. The cytotoxicity of EPS to the model cell line THP-1 was examined. Figure 1A reveals that 0.1–200 μg/mL EPS did not significantly affect the viability of THP-1 cells, suggesting that EPS is not toxic to the cell line at concentrations between 0.1–200 μg/mL. However, the addition of LPS did affect cell growth. As shown in Figure 1B, cotreatment of THP-1 cells with 1 μg/mL LPS and 20–200 μg/mL EPS caused a 10–20% decrease in the number of viable cells.

Theoretically, as a control in the functional study of EPS, a strain of *B. amyloliquefaciens* that does not produce exopolysaccharides should be cultured simultaneously, and the resulting medium being subjected to the same isolation protocol used for EPS preparation. However, we do not know any strain of *B. amyloliquefaciens* that does not produce exopolysaccharides. Thus, the control for EPS was prepared by incubating the same volume of sterile medium at 37 °C for 48 h, mimicking the condition of amy-1 cultivation except that the medium was not inoculated with any micro-organisms. The medium was then subjected to an isolation protocol, the same as that used for preparing EPS. The resulting powders, abbreviated as PPT (please see Section 4.3), with total carbohydrates and total proteins being 52.00 ± 3.10% and 10.95 ± 0.64% of the total mass, respectively, were used as a control of EPS in experiments. The cytotoxicity of PPT was also analyzed. Figure 1C shows that 0.1–100 μg/mL PPT did not significantly affect the viability of THP-1 cells, but 150 and 200 μg/mL of PPT exhibited 10–20% inhibition on cell growth. In Figure 1D, the combined effects of 1 μg/mL LPS and 10–200 μg/mL PPT were analyzed. Only LPS plus 200 μg/mL PPT displayed approximately 10% suppression on cell growth. Therefore, the effects of EPS and PPT on cell growth are different.

The anti-inflammatory effect of EPS was tested in LPS-treated THP-1 cells. Figure 2A shows that LPS increased the expression of inducible nitric oxide synthase (iNOS; Lane 2), whereas the presence of 10, 50, or 100 μg/mL EPS obviously inhibited this effect of LPS in a dose-dependent manner (Lanes 3, 4, and 5). Similarly, LPS increased the expression of cyclooxygenase-2 (COX-2; Figure 2B, Lane 2), and this reaction was also dose-dependently suppressed by 10–100 μg/mL EPS (Figure 2B, Lanes 3, 4, and 5). The effect of PPT on iNOS and COX-2 expression was also examined. Consequently, unlike EPS, 10–100 μg/mL PPT did not suppress LPS-induced iNOS (Figure 2C, Lanes 3, 4, and 5 vs. Lane 2) and COX-2 expression (Figure 2D, Lanes 3, 4, and 5 vs. Lane 2), suggesting that PPT does not have anti-inflammatory activity.

Furthermore, LPS increased the production of TNF-α (Figure 2E, Group 2) and IL-6 (Figure 2F, Group 2) from THP-1 cells; yet the addition of EPS significantly decreased TNF-α (Figure 2E, Group 3) and IL-6 (Figure 2F, Group 3) production. Subsequently, the phagocytic activity of LPS-, LPS + EPS-, or EPS-treated THP-1 cells was analyzed. LPS increased the phagocytic activity of THP-1 cells as expected (Figure 2G, Group 2 vs. Group 1), whereas EPS alone increased the phagocytic activity of the cells even further than LPS did (Figure 2G, Group 3 vs. Group 2). However, cotreatment of THP-1 cells with both LPS and EPS (Figure 2G, Group 4) reduced the phagocytic activity of THP-1 cells as compared to LPS treatment (Figure 2G, Group 2). Therefore, when LPS and EPS are added together, the phagocytic activity of THP-1 cells is suppressed compared to LPS-treated cells, but EPS alone can increase the phagocytic activity of the cells. Together, data in Figure 2 demonstrate that EPS diminishes LPS-induced proinflammatory biomarkers in THP-1 cells. Thus, the anti-inflammatory activity of EPS was then tested in vivo.

Figure 3 displays that 12-*O*-tetradecanoylphorbol-13-acetate (TPA) stimulation caused ear edema in mice at 4, 16, and 24 h after TPA application (Group 2). EPS treatment (250, 500, and 750 μg/ear; Groups 4, 5, and 6, respectively) resulted in dose-dependent amelioration of the ear edema in mice at these time points. The effects of 500 and 750 μg/ear of EPS were close and were both similar to that of 500 μg/ear indomethacin (Group 3), an anti-inflammatory medicine. These data demonstrate that EPS has an obvious anti-inflammatory effect in vivo.

### 2.2. Molecular Mechanism—The Inhibitor Kappa B Kinase (IKK)/Nuclear Factor-κB (NF-κB) Pathway

LPS activates the proinflammatory IKK/NF-κB pathway [24,25]. Thus, whether EPS inhibits this pathway was examined in our study. As shown in Figure 4A, LPS obviously increased IKK phosphorylation (Lane 2); the addition of EPS decreased the level of IKK phosphorylation (Lane 3). Consistently, in Figure 4B, phosphorylation of the inhibitor of NF-κB (IκB) was elevated by LPS (Lane 2); the presence of EPS reduced the level of phosphorylated IκB (Lane 3). Furthermore, Figure 4C shows that LPS increased the level of nuclear p65, a subunit of NF-κB (Lane 2), whereas the addition of EPS obviously diminished the nuclear presence of p65 (Lane 3), indicating that EPS inhibited the nuclear translocation of NF-κB induced by LPS. Taken together, data in Figure 4 suggest that EPS inhibits the LPS-activated IKK/NF-κB pathway.

### 2.3. Molecular Mechanism—The Roles of Mitogen-Activated Protein Kinases (MAPKs)

MAPKs are also activated by LPS through the activation of toll-like receptor 4 [26]. Figure 5A shows that LPS activated p38 (Lane 2). When EPS was also present, p38 activation appeared to be inhibited by 10 and 50 μg/mL EPS, whereas an EPS dose-dependent recovery of p38 activation was observed (Lanes 3, 4, and 5). The phosphorylation level of p38 increased with increasing EPS concentrations. At 100 μg/mL EPS (Lane 5), the p38 phosphorylation level was no different compared to LPS-induced p38 activation (Lane 2). LPS induced the activation of c-Jun-N-terminal kinase 2 (JNK; Figure 5B, Lane 2), yet the presence of 10–100 μg/mL EPS did not suppress nor further increase JNK phosphorylation (Lanes 3, 4, and 5; Figure 5B). Namely, EPS did not influence the effect of LPS on JNK. LPS activated the extracellular signal-regulated kinase 1/2 (ERK1/2; Figure 5C, Lane 2); the addition of 10–100 μg/mL EPS dose-dependently suppressed ERK1/2 activation (Lanes 3, 4, and 5; Figure 5C). Therefore, the effects of EPS on the three MAPKs were different in LPS-treated cells. Hence, we further analyzed the effects of EPS alone on MAPKs. EPS alone activated p38 in a dose-dependent manner (Figure 5D, Lanes 2–6), but did not obviously activate or inhibit JNK (Figure 5E, Lanes 2–6). However, EPS dose-dependently inhibited ERK1/2 phosphorylation (Figure 5F, Lanes 2–6). For comparison, the effects of PPT on MAPKs were also examined. As shown in Figure 5G–I, 0.1–100 μg/mL PPT (Lanes 2–6) did not significantly activate or inhibit p38 (G), JNK (H), or ERK1/2 (I). That is, PPT does not have an obvious effect on any of the MAPKs.

Thus, EPS inhibits ERK1/2 and has no effect on JNK, but activates p38, and p38 is known to promote inflammation through the activation of activator protein 1 (AP-1), which promotes the expression of inflammatory cytokines [27]. Nonetheless, p38 also activates the transcription factor nuclear factor erythroid 2–related factor 2 (Nrf2), which can counteract the action of NF-κB [27]. Hence, we tested the role of p38. The addition of a p38 inhibitor, SB202190, further decreased the LPS-induced production of TNF-α (Figure 2E, Group 4) and IL-6 (Figure 2F, Group 4) as compared to the inhibitory effect of EPS (Group 3; Figure 2E,F), indicating that p38 was involved in promoting the production of these proinflammatory cytokines in LPS and EPS cotreated cells, and the p38 inhibitor repressed this function of p38. This is consistent with the known proinflammatory effect of p38. In Figure 4A,B, the p38 inhibitor did not obviously affect the levels of IKK phosphorylation and IκB phosphorylation that were already inhibited by EPS (Lane 4 vs. Lane 3). However, in Figure 4C, NF-κB nuclear translocation that was suppressed by EPS (Lane 3) was re-boosted by the addition of SB202190 (Lane 4), suggesting that p38 played a role in inhibiting NF-κB nuclear translocation. The results in Figure 4 are consistent with the notion that p38 activates Nrf2, which hinders the action of NF-κB, but does not act on IKK and IκB [27]. Therefore, whether EPS-activated Nrf2 was characterized subsequently.

### 2.4. Molecular Mechanism—The Anti-Oxidative Pathway

Nrf2 induces the expression of anti-oxidative enzymes, including heme oxygenase-1 (HO-1) and the glutamate–cysteine ligase modifier subunit (GCLM) [27]. Therefore, whether EPS enhanced the expression of these enzymes and Nrf2 was examined. Figure 6A shows that LPS elevated the expression of HO-1 (Lane 2) as expected [27]; EPS alone (Lane 3) or EPS plus LPS (Lane 4) also promoted the expression of HO-1. The result of Lane 3 suggests that EPS promotes HO-1 expression. The same assay was performed, using PPT. As a result, Figure 6B reveals that HO-1 expression level was not significantly increased by PPT (Lane 3) compared to the control (Lane 1). In cells treated by both PPT and LPS, HO-1 expression was enhanced as expected due to the presence of LPS (Lane 4). Thus, PPT does not activate the expression of HO-1. Subsequently, Figure 6C,D exhibit that LPS increased the expression of GCLM (C, Lane 2) and Nrf2 (D, Lane 2). Similarly, EPS alone, or EPS plus LPS, also promoted the expression of GCLM (C, Lanes 3 and 4) and Nrf2 (D, Lanes 3 and 4). Overall, Figure 6 suggests that EPS, like LPS, can activate the Nrf2/HO-1 anti-oxidative pathway.

Subsequently, when p38 inhibitor was added, the expression of HO-1, GCLM, and Nrf2 was obviously decreased in LPS and EPS cotreated cells (Figure 6A,C,D, Lane 5), as well as in LPS and PPT cotreated cells (Figure 6B, Lane 5), implying that p38 was likely involved in the activation of the Nrf2/HO-1 pathway in these cells. To further assess the role of p38 in EPS-induced activation of the Nrf2/HO-1 pathway, Figure 7A shows that EPS alone activated p38 and increased the expression of Nrf2, HO-1, and GCLM (Lane 2), yet these reactions were all obviously inhibited by the p38 inhibitor (Lane 3), suggesting that p38 mediated EPS-induced activation of the Nrf2/HO-1 pathway. EPS does not inhibit the LPS-induced activation of JNK. Thus, whether JNK is involved in the activation of the Nrf2/HO-1 pathway in EPS and LPS cotreated cells was also examined. Figure 7B demonstrates that in cells treated with LPS or LPS + EPS, JNK was activated (Lanes 2 and 5, respectively) but was not in those treated with EPS alone (Lane 4). Moreover, the expression levels of HO-1, GCLM, and Nrf2 were enhanced with treatment by LPS (Lane 2), EPS (Lane 4), or LPS + EPS (Lane 5). These results are consistent with those of Figure 5 and Figure 6. When a JNK inhibitor was added to cells treated with LPS or LPS + EPS, JNK activation was indeed suppressed (Figure 7B, Lanes 3 and 6, respectively). However, the expression levels of HO-1, GCLM, and Nrf2 were not obviously affected by the inhibitor (Figure 7B, Lanes 3 and 6). It suggests that JNK is not important in the activation of the Nrf2/HO-1 pathway in cells treated with LPS or LPS + EPS.

The anti-oxidative effect of EPS was then examined through analysis of the intracellular level of reactive oxygen species (ROS). As shown in Figure 7C, LPS apparently enhanced the level of intracellular ROS (Group 2), and the addition of EPS significantly reduced the amount of ROS (Group 3). However, in the presence of p38 inhibitor, the ROS level was increased again (Group 4 vs. Group 3). Figure 7C supports the proposal that EPS activates p38, which in turn activates the Nrf2/HO-1 pathway, resulting in a reduced ROS level.

## 3. Discussion

In this study, the exopolysaccharide preparation of *B. amyloliquefaciens* amy-1 was shown to impede the proinflammatory actions of LPS on THP-1 cells including: LPS-induced expression of iNOS, COX-2, TNF-α, and IL-6, phagocytic activity of the cells, and activation of the IKK/NF-κB pathway. In the animal tests, EPS effectively ameliorated ear inflammation in mice, and the efficacy of EPS was similar to that of the clinical anti-inflammatory medicine indomethacin. These results suggest that EPS possesses anti-inflammatory activity. Our study further indicates that the underlying mechanisms are associated with several intracellular signaling pathways. EPS inhibits LPS-induced activation of the IKK/NF-κB pathway and that of ERK1/2, yet at the mean time activates p38 and the Nrf2/HO-1 pathway. The former reduces the intensity of inflammation caused by LPS; the latter enhances anti-oxidative function of the cells that decreases oxidative stress caused by inflammation, protects cells from oxidative damages, and also reduces the intensity of inflammation. Moreover, we found that JNK is likely not involved in activating the Nrf2 pathway in THP-1 cells.

The compositions of EPS and PPT are obviously different, because the carbohydrate content of EPS is over 95% of its total mass, whereas PPT contains 52% carbohydrates and a substantial amount of proteins (over 10%). PPT was likely composed of ethanol-insoluble components of the MRS broth. The effect of PPT was also tested to explore whether medium ingredients might have contributed to the observed activity of EPS. As a result, EPS and PPT exhibited distinct effects on the growth of THP-1 cells. PPT mildly yet significantly inhibited the growth of THP-1 cells at 150 and 200 μg/mL, yet EPS did not. It is speculated that at higher concentrations the content of PPT might have significantly affected the osmotic pressure of the culture medium of THP-1 cells, leading to the observed dose-dependent effect of PPT on the survival of THP-1 cells. Moreover, unlike EPS, PPT did not show any anti-inflammatory activity. PPT had no effect on suppressing LPS-induced iNOS and COX-2 expression, did not inhibit nor activate MAPKs, and neither promoted HO-1 expression, suggesting that the ethanol-insoluble components of the medium have no anti-inflammatory activity. Taken together, we conclude that the activity of EPS was not contributed by medium ingredients.

EPS demonstrated differential effects on MAPKs. EPS alone activated p38 and inhibited ERK1/2 in a dose-dependent manner, but did not activate, nor inhibit JNK. LPS was reported to activate ERK1/2, JNK, and p38 [28], and our data also confirmed this point. However, in LPS and EPS cotreated cells, ERK was inhibited, but JNK was activated, suggesting that EPS hindered LPS-induced activation of ERK1/2, but not that of JNK. The latter coincided with the finding that EPS had no effect on JNK. The reaction of p38 in LPS and EPS cotreated cells was different. In the presence of 10 and 50 μg/mL EPS, p38 activation was obviously decreased compared to the control (LPS-treated cells), suggesting that EPS inhibited LPS-induced p38 activation. However, the level of p38 activation actually increased with increasing concentrations of EPS in LPS and EPS cotreated cells (Figure 5A), indicating that EPS on one hand impeded the action of LPS on activating p38, whereas on the other hand activated p38 by itself. The former again suggests that EPS prohibits the effect of LPS, and the latter agrees with the finding that p38 was activated in cells treated with EPS alone. Therefore, we propose that EPS can activate p38, yet in the meantime inhibits the effect of LPS on p38. Thus, the observed increasing levels of p38 activation with increasing concentrations of EPS in Figure 5A Lanes 3–5 was due to the activation of p38 by EPS. We have not found any other anti-inflammatory compound in the literature reported to behave this way on p38 when cotreated with LPS.

ERK1/2 [29,30], p38 [31,32,33], and JNK [33,34], have all been suggested to activate the Nrf2 pathway. However, our data suggest that JNK is likely not involved or not important in the activation of the Nrf2/HO-1 pathway in THP-1 cells. In LPS and EPS cotreated cells, ERK1/2 was inhibited, whereas p38 and JNK were activated. However, inhibiting JNK in these cells did not obviously affect the expression of Nrf2, HO-1, and GCLM, but suppressing p38 effectively inhibited the expression of these factors. Theoretically, if JNK and p38 both activate Nrf2, repressing either one of them should not effectively impede the activation of the Nrf2/HO-1 pathway since the other kinase can still activate the pathway. However, in our data, p38 inhibitor efficiently hindered the activation of the Nrf2/HO-1 pathway in LPS + EPS-treated cells, but JNK inhibitor did not. Hence, our data suggest that p38 plays a crucial role in activating Nrf2 in THP-1 cells, but JNK does not. In previous studies JNK was indicated to activate the Nrf2 pathway in RAW264.7 cells [33,34]; however, other studies showed that the downregulation of JNK was accompanied with the upregulation of Nrf2 in a microglial cell line [35], and in HepG2 cells [36]. Therefore, the role of JNK on regulating Nrf2 seems to be distinct between different cell lines or tissues.

Therefore, our proposed mechanisms underlying the anti-inflammatory effect of EPS are depicted in Figure 8. LPS activates the IKK/NF-κB pathway and the MAPKs p38, ERK1/2, and JNK. All MAPKs activate AP-1 [27]. Together, the IKK/NF-κB pathway and the MAPKs promote inflammation. Additionally, ERK1/2 and p38 activate the Nrf2/HO-1 pathway, which activates the expression of anti-oxidative factors and thus reduces oxidative stress. The activation of the anti-oxidative pathway by MAPKs may be a protective strategy exerted by the cell to reduce self-injury, or to avoid excessive inflammatory responses [37]. EPS inhibits LPS-induced IKK activation, leading to the suppression of the IKK/NF-κB signaling and resulting in a reduced inflammation intensity. Furthermore, LPS-induced MAPK activation is partially hindered by EPS. EPS inhibits ERK1/2 activation and hinders LPS-induced ERK1/2 activation. However, EPS does not have an obvious effect on JNK and does not interfere with LPS-induced JNK activation. EPS activates p38 but inhibits the action of LPS on p38. Consequently, JNK and p38 still activate some proinflammatory reactions through AP-1. Hence, in our data, EPS mitigated but did not completely prohibit LPS-induced TNF-α and IL-6 production (Figure 2E,F, Group 3), yet adding a p38 inhibitor further reduced the levels of these cytokines (Figure 2E,F, Group 4). Nonetheless, p38 also activates Nrf2, which reduces inflammation-caused oxidative stress and interferes with NF-κB activity due to the fact that Nrf2 activates the expression of anti-oxidative proteins and can counteract the action of NF-κB [27,38]. Thus, EPS can protect cells from oxidative damages and decrease the intensity of inflammation. The mechanism by which EPS can differentially regulate MAPKs is unclear; furthermore, the mechanism through which EPS activates p38 yet simultaneously inhibits p38 activation by LPS is intriguing. Further investigation is required to answer these questions.

That EPS contains anti-inflammatory activity which coincides with the finding that the extracellular extracts of *B. amyloliquefaciens* MBMS5 contained anti-inflammatory function [15]. Other probiotic exopolysaccharides suggested to have anti-inflammatory activity include those produced from *Lactobacillus paraplantarum* [39], *Bacillus subtilis* [40,41], *Lactobacillus reuteri* [16], *Lactococcus lactis* [42], *Bifidobacterium longum* [43], etc. However, the molecular mechanisms underlying the anti-inflammatory effects of these exopolysaccharides are mostly unclarified, except that *B. subtilis* exopolysaccharides were reported to inhibit NF-κB expression [41], or to induce M2 phenotype of macrophages [40]. This study provides a clearer picture for the potential molecular mechanisms mediating the anti-inflammatory effect of the exopolysaccharides of a probiotic.

Treating THP-1 cells with EPS alone obviously increased the phagocytic activity of the cells (Figure 2G, Group 3). In addition to being a biomarker in inflammation, increased phagocytic activity of neutrophils or macrophages is also a sign of enhanced innate immunity. For example, polysaccharides isolated from *Ganoderma lucidum* were suggested to increase immunity, and one of the evidences was that they increased the phagocytic activity of macrophages [44]. Polysaccharides isolated from *Angelica dahurica* were suggested to have immunomodulatory properties and were shown to enhance the phagocytic activity of RAW264.7 cells [45]. Therefore, EPS may have immunomodulatory activity, which deserves to be further investigated.

In our previous study, EPS was found to activate bitter-taste receptors in enteroendocrine cells [22]. More and more evidence has suggested that bitter-taste receptors are associated with immune modulation [23,24,46]. Therefore, whether EPS activates bitter-taste receptors in THP-1 cells, and whether this activation is associated with the anti-inflammatory activity of EPS require further investigations. On the other hand, LPS binds to toll-like receptor 4 [28], whereas our data reveal that EPS counteracts the pro-inflammatory actions of LPS. Hence, it is reasonable to speculate that EPS may work as an antagonist of toll-like receptor 4, resulting in inhibition on the effects of LPS. Yet, LPS-induced JNK activation was not affected by EPS (Figure 8). Put differently, EPS did not inhibit all the effects of LPS that are mediated by activated toll-like receptor 4. Therefore, whether EPS binds to toll-like receptor 4 needs to be carefully evaluated.

In conclusion, the exopolysaccharides of *B. amyloliquefaciens* amy-1 possess anti-inflammatory activity. The underlying mechanisms are associated with the inhibition of the IKK/NF-κB pathway and ERK1/2, and the activation of p38 followed by that of the Nrf2/HO-1 pathway. Together, these mechanisms reduce the severity of inflammation. These findings support the proposal that exopolysaccharides are a crucial element in the anti-inflammatory functions of at least some probiotics.

## 4. Materials and Methods

### 4.1. Cell Culture, Treatments, and Cytotoxicity Assay

THP-1 cells were purchased from Bioresource Collection and Research Center (Hsinchu, Taiwan) and cultured in antibiotic-free RPMI 1640 medium containing 10% fetal bovine serum at 37 °C in a humidified incubator supplied with 5% CO_2_. To perform EPS functional assays, the cells were seeded into 35-mm culture dishes (2 × 10^6^/well), and cultured in a serum-free, antibiotic-free medium containing 0.2% bovine serum albumin and 100 ng/mL TPA for 24 h to allow cells to differentiate into M0-state macrophage-like cells [47]. Then, the cells were treated with vehicle (the same volume of sterile distilled water), 1 μg/mL LPS (dissolved in sterile distilled water), and/or a desired concentration of EPS for the indicated duration. In experiments using a kinase inhibitor, cells were pretreated with 20 μM of the inhibitor for 30 min, followed by the addition of LPS and/or EPS. For cytotoxicity assays, cells were seeded in a 96-well plate and treated with TPA to allow differentiation into M0-state cells as aforementioned. The cells were then treated with the indicated concentration of LPS, EPS, and/or PPT in a serum-free medium for 24 h. After washing the cells with phosphate-buffered saline (PBS, pH 7.4), a Cell Counting Kit-8 reagent (Target Molecule Corp., Boston, MA, USA) was added into the medium, and the cells were incubated for 4 h as per the supplier’s instructions. Absorbance at 450 nm was then measured by using Varioskan LUX multimode microplate reader (Thermo Fisher Scientific Inc., Waltham, MA, USA). Cell viability relative to the control (cells treated with vehicle) was calculated.

### 4.2. Antibodies and Chemicals

An antibody against iNOS was purchased from Novus Biologicals (Centennial, CO, USA). The primary antibodies targeting COX-2, the IKKα, IKKβ, phosphorylated IKKα (Ser176)/IKKβ (Ser177), NF-κB p65 subunit, IκBα, phosphorylated IκBα (Ser32/36), p38, phosphorylated p38 (Thr180/Tyr182), JNK, phosphorylated JNK (Thr183/Tyr185), ERK1/2, phosphorylated ERK1/2 (Thr202/Tyr204), HO-1, and Nrf2 were purchased from Cell Signaling Technology (Beverley, MA, USA). SP600125 and an antibody against actin were purchased from Calbiochem (Merck Millipore, Darmstadt, Germany). Antibodies for Lamin B1, α-tubulin, GCLM, and SB202190 were bought from Abcam (Cambridge, UK). The secondary antibody mouse anti-rabbit IgG conjugated with horseradish peroxidase (HRP) was purchased from Santa Cruz Biotechnology (Dallas, TX, USA). The secondary antibody goat anti-mouse IgG antibody conjugated with HRP, protease inhibitor cocktail, and phosphatase inhibitor cocktail were from Merck Millipore. Other chemicals were ordered from Sigma-Aldrich (St. Louis, MO, USA).

### 4.3. Preparation and Analysis of EPS and the Control (PPT)

Exopolysaccharides were isolated from the culture medium (MRS broth) of *B. amyloliquefaciens* amy-1 through ethanol precipitation as described previously [20]. The isolated exopolysaccharides (i.e., EPS) were lyophilized, stored at −20 °C, and dissolved in sterile distilled water in appropriate concentrations for experiments. The exopolysaccharide solution was subjected to total carbohydrate analysis, total protein analysis, total lipid analysis, and total polyphenol analysis (N = 3) as described previously [20].

The control was prepared by incubating the same volume of sterile MRS broth without inoculating any micro-organism in a shaking incubator at 37 °C for 48 h like the cultivation of amy-1. The medium was then subjected to ethanol precipitation by the same protocol used for preparing EPS. The resulting precipitate (PPT) was collected, lyophilized, dissolved in sterile distilled water, and assayed for total carbohydrates and total proteins. In experiments, the vehicle control for EPS or PPT was the same volume of sterile distilled water.

### 4.4. Protein Extraction and Western Blotting

THP-1 cells were cultured and treated as aforementioned. Cells were incubated for 16 h for the analysis of iNOS and COX-2; 10 min for that of IKK and IκB; 90 min for nuclear p65; 1 h (Figure 5) or 24 h (Figure 7) for MAPKs; and 24 h for HO-1, GCLM, and Nrf2. Cells were then washed twice with PBS and lysed through submersion in a proper volume of RIPA lysis buffer (Merck Millipore) containing a protease inhibitor cocktail and a phosphatase inhibitor cocktail. Lysed cells were scraped off the plate on ice and centrifuged. The supernatant was collected and subjected to Bradford Assay for determining protein concentrations. For nuclear NF-κB analysis, nuclear proteins were extracted using a Nuclear Extraction kit (Merck Millipore) following the manufacturer’s instructions.

Equal amounts of proteins were sampled from each treatment, subjected to acrylamide gel electrophoresis, and then transferred onto Immobilon-P Membrane PVDF membrane (Merck Millipore). After blocking the membranes with BlockPRO Protein-Free Blocking Buffer (Energenesis Biomedical Co., Taipei, Taiwan) for 1 h at room temperature, they were hybridized with primary antibodies diluted (2000–10,000 folds) in Immobilon Signal Enhancer for Immunodetection (Merck Millipore) at 4 °C overnight. Then, they were washed at least thrice with Tris-buffered saline (pH 7.4) containing 0.1% Tween-20 (TBS-T), and hybridized with HRP-conjugated secondary antibodies for 1 h at room temperature. Subsequently, they were washed thoroughly with TBS-T, and a proper volume of SuperSignal West Pico PLUS Chemiluminescent Substrate (Thermo Fisher Scientific), or Clarity Max Western ECL Substrate (Bio-Rad Laboratories, Hercules, CA, USA), was added to the membranes. Signals were detected using an imaging system (ChemiDoc XRS+ Imaging Systems, Bio-Rad), and band intensities were analyzed using the supplied software (Image Lab, Bio-Rad).

### 4.5. Enzyme-Linked Immunosorbent Assay (ELISA) for TNF-α and IL-6

1 × 10^6^ THP-1 cells were seeded into 35 mm culture dishes and differentiated to M0-state macrophage-like cells by TPA as aforementioned. Then, they were washed twice with PBS and treated with SB202190 (Group 4 of Figure 2E,F) or vehicle (Groups 1, 2, and 3 of Figure 2E,F) for 30 min, followed by the addition of 1 μg/mL LPS and/or 100 μg/mL EPS, as indicated in the figures, and incubated for 16 h. The medium was collected and analyzed for TNF-α and IL-6 concentrations by using the respective ELISA kits (Invitrogen, Thermo Fisher Scientific) as per the manufacturer’s instructions.

### 4.6. Phagocytosis Assay

THP-1 cells were seeded into 96-well culture plates (1 × 10^4^ cells/well) and differentiated to M0 state by using TPA. After washing the cells twice with PBS, they were treated with 1 μg/mL LPS and/or 100 μg/mL EPS for 24 h. Then, they were washed twice with PBS, incubated in PBS containing 40 μg/mL neutral red for 1 h at 37 °C, and washed twice with PBS again. Subsequently, the cells were incubated in a lysis solution (50% acetic acid and 50% ethanol, *v*/*v*) for 1 h at room temperature. Absorbance at 492 nm was then measured using a microplate reader (Varioskan™ LUX multimode microplate reader, Thermo Fisher Scientific).

### 4.7. Animal Tests

Eight-week-old male ICR mice were purchased from BioLASCO (Taipei, Taiwan) and allowed 2-week acclimation before the experiments. The mice were housed under a 12 h light–dark cycle, allowed free access to water and food, and fed with a regular laboratory rodent diet. The mouse ear edema assay was conducted as described previously [48], with modifications. The mice were randomly divided into six groups of five. TPA (3 μg/ear dissolved in 20 μL acetone) was applied on the right ears of the mice in Groups 2–6, and 20 μL acetone was applied to the right ears of the mice in Group 1. One hour later, 500 μg/ear indomethacin (dissolved in 20 μL of acetone) was applied to the right ears of the mice in Group 3; 250, 500, and 750 μg/ear EPS (dissolved in 20 μL WA solvent, which was 50% water and 50% acetone) was applied to the right ears of the mice in Groups 4, 5, and 6, respectively. For mice in Groups 1 and 2, 20 μL of WA solvent was applied to the right ears. The thickness of the treated ears was measured using a dial thickness gauge (Peacock, Ozaki, Tokyo, Japan) before stimulation and 4, 16, and 24 h after TPA stimulation.

### 4.8. Intracellular ROS Assay

THP-1 cells were seeded into 35-mm culture dishes (1 × 10^6^ cells/dish) and differentiated to the M0 state by using TPA. The cells were then treated with serum-free media containing 100 μg/mL of EPS for 1 h. After washing the cells twice with PBS, they were incubated in serum-free media containing 10 μM dichlorofluorescin diacetate and 20 μM SB202190 (or vehicle) for 30 min. The cells were washed with PBS again and treated with serum-free media containing 1 μg/mL LPS (or vehicle) for 1 h. Then, they were washed twice with PBS and scraped off the dish under PBS. The resulting suspension was analyzed for fluorescence intensity (excitation wavelength 504 nm and emission wavelength 529 nm) in a fluorescence detector (Modulus Single Tube Multimode Reader, Turner BioSystems, Sunnyvale, CA, USA).

### 4.9. Statistical Analysis

Data were analyzed through one-way analysis of variance followed by Scheffe’s post hoc test by using the Microsoft program Excel. Significant difference was identified when *p* < 0.05.

## Figures and Tables

**Figure 1 ijms-23-10237-f001:**
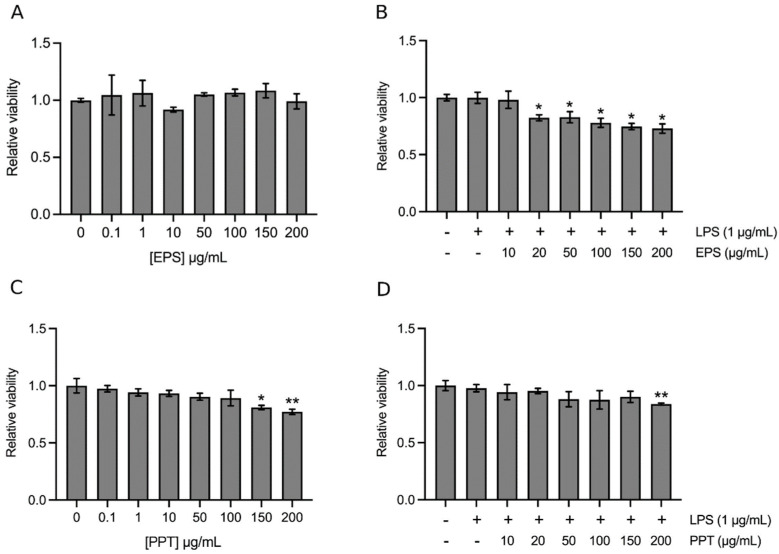
Analyses of EPS (the exopolysaccharide preparation of amy-1) and PPT (precipitate of the medium without inoculating any micro-organisms) cytotoxicity on THP-1 cells. A and C, THP-1 cells were treated with vehicle (the same volume of sterile distilled water), 0.1–200 μg/mL EPS (**A**), or 0.1–200 μg/mL PPT (**C**) for 24 h. (**B**,**D**) cells were treated with vehicle (the same volume of sterile distilled water), 1 μg/mL LPS alone, LPS and 10–200 μg/mL EPS (**B**), or LPS and 10–200 μg/mL PPT (**D**) for 24 h. Cell viability relative to the control (vehicle) was calculated. Data are presented as means ± standard deviation of three independent experiments, each in duplicate. * *p* < 0.05, ** *p* < 0.005 vs. the control.

**Figure 2 ijms-23-10237-f002:**
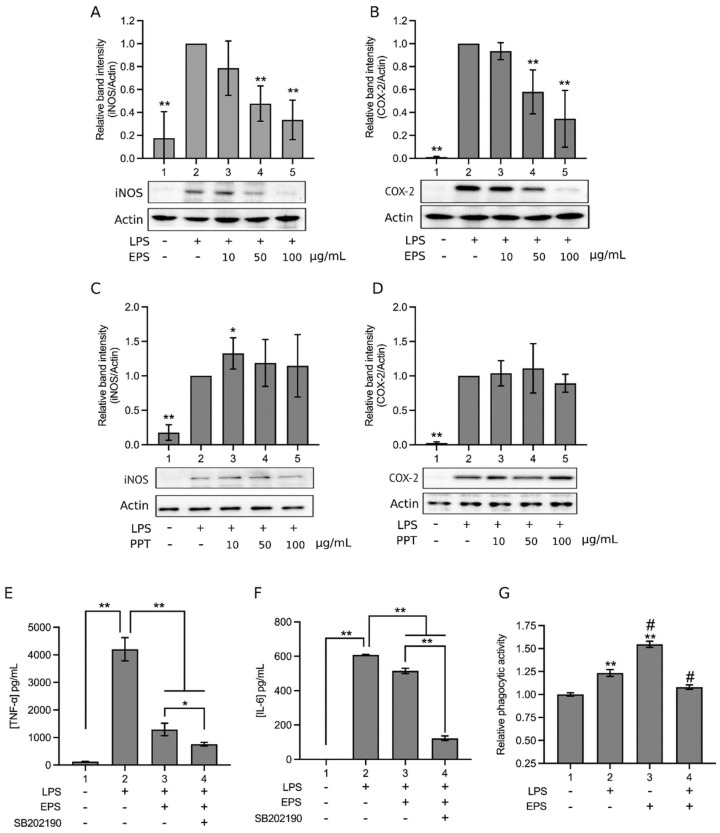
EPS reduced inflammatory biomarker levels in lipopolysaccharide (LPS)-treated THP-1 cells. (**A**,**B**) THP-1 cells were treated with vehicle, 1 μg/mL LPS, or LPS and 10, 50, or 100 μg/mL EPS for 16 h. The expression levels of inducible nitric oxide synthase (iNOS; **A**) and cyclooxygenase-2 (COX-2; **B**) in the cells were analyzed, using Western blotting. (**C**,**D**) the same experiments of (**A**,**B**) were performed, but EPS was replaced by PPT. Relative band intensity against Lane 2 was determined after normalization by actin. Data are presented as means ± standard deviations of four independent experiments. * *p* < 0.05, ** *p* < 0.005 vs. Lane 2. (**E**,**F**) THP-1 cells were treated with vehicle (Group 1), 1 μg/mL LPS (Group 2), LPS and 100 μg/mL EPS (Group 3), or LPS, EPS, and 20 μM SB202190 (a p38 inhibitor; Group 4) for 16 h. The culture media were subjected to the concentration analyses of tumor necrosis factor-α (TNF-α; **E**) and interleukine-6 (IL-6; **F**) by using ELISA. Data are presented as the mean ± standard deviation of an experiment in triplicate. * *p* < 0.05 and ** *p* < 0.005 between the indicated groups. (**G**) THP-1 cells were treated with vehicle (Group 1), 1 μg/mL LPS (Group 2), 100 μg/mL EPS (Group 3), or both LPS and EPS (Group 4) for 24 h, and the phagocytic activities of the cells were then analyzed. Data are presented as the mean ± standard deviation of an experiment in triplicate. ** *p* < 0.005 vs. Group 1; # *p* < 0.05 vs. Group 2.

**Figure 3 ijms-23-10237-f003:**
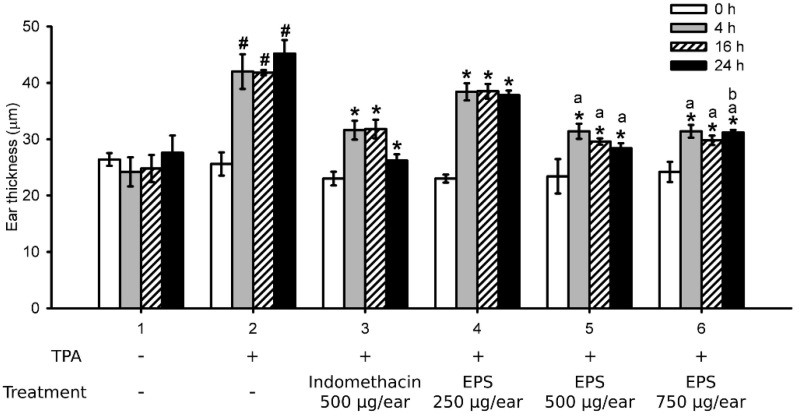
EPS ameliorated 12-*O*-tetradecanoylphorbol-13-acetate (TPA)-induced ear edema in animal models. Mice were treated with TPA (Groups 2–6) or vehicle (Group 1) on their right ears. One hour later, vehicle (Groups 1 and 2), 500 μg/ear indomethacin (Group 3), or 250, 500, or 750 μg/ear EPS (Groups 4, 5, and 6, respectively) was applied on the ear. Ear thickness was measured before (0 h) and at 4, 16, and 24 h after TPA treatment. Data are presented as the mean ± standard deviation of each group (N = 5). # *p* < 0.005 vs. 0 h of Group 2; * *p* < 0.005 vs. the same time point of Group 2; ^a^
*p* < 0.005 vs. the same time point of Group 4; ^b^
*p* < 0.005 vs. the same time point of Group 5.

**Figure 4 ijms-23-10237-f004:**
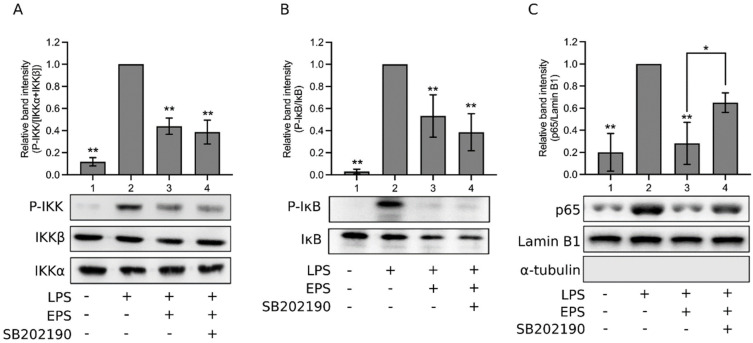
EPS inhibited the inhibitor kappa B kinase (IKK)/nuclear factor-κB (NF-κB) pathway. THP-1 cells were treated with vehicle (Lane 1), 1 μg/mL LPS (Lane 2), LPS and 100 μg/mL EPS (Lane 3), or LPS, EPS, and 20 μM SB202190 (Lane 4) for 10 min (**A**,**B**) or 90 min (**C**). The levels of total IKK (IKKα + IKKβ) and phosphorylated IKK (**A**), total IκB (the inhibitor of NF-κB) and phosphorylated IκB (**B**), nuclear p65 subunit, as well as lamin B1 and α-tubulin (**C**), were assayed using Western blotting. Relative band intensity (normalized) vs. Lane 2 was determined. Data are presented as the mean ± standard deviation of four independent experiments. * *p* < 0.05 and ** *p* < 0.005 vs. Lane 2 or between the indicated groups. No significant difference was found between Lanes 3 and 4 of (**A**), and between those of (**B**).

**Figure 5 ijms-23-10237-f005:**
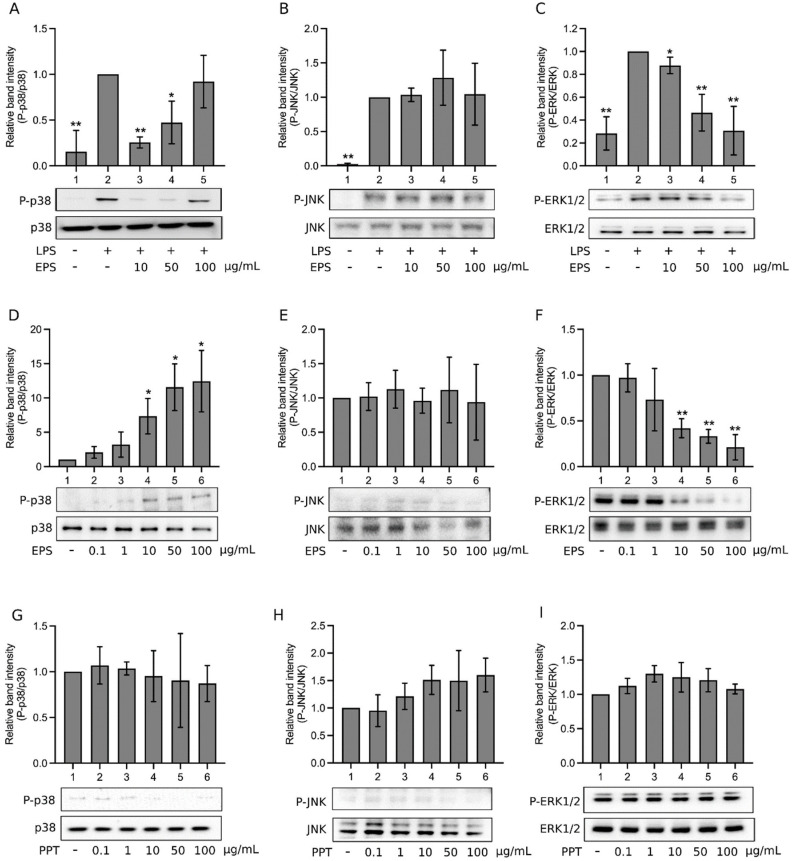
The effects of EPS on mitogen-activated protein kinases (MAPKs). (**A**–**C**) THP-1 cells were treated with vehicle (Lane 1), 1 μg/mL LPS (Lane 2), or LPS and 10, 50, or 100 μg/mL EPS (Lanes 3–5) for 1 h. (**D**–**F**) THP-1 cells were treated with vehicle (Lane 1) or 0.1, 1, 10, 50, or 100 μg/mL EPS (Lanes 2–6) for 1 h. (**G**–**I**) the same experiments of (**D**–**F**) were performed, but EPS was replaced by PPT. The levels of phosphorylated p38 and total p38 (**A**,**D**,**G**), phosphorylated c-Jun-N-terminal kinase 2 (JNK) and total JNK (**B**,**E**,**H**), phosphorylated extracellular signal-regulated kinase 1/2 (ERK1/2) and total ERK1/2 (**C**,**F**,**I**) were assayed, using Western blotting. Normalized band intensity relative to Lane 2 (**A**–**C**) or Lane 1 (**D**–**I**) was determined. Data are presented as the mean ± standard deviation of three (**A**–**F**) or four (**G**–**I**) independent experiments. * *p* < 0.05 and ** *p* < 0.005 vs. Lane 2 (**A**–**C**) or Lane 1 (**D**–**I**).

**Figure 6 ijms-23-10237-f006:**
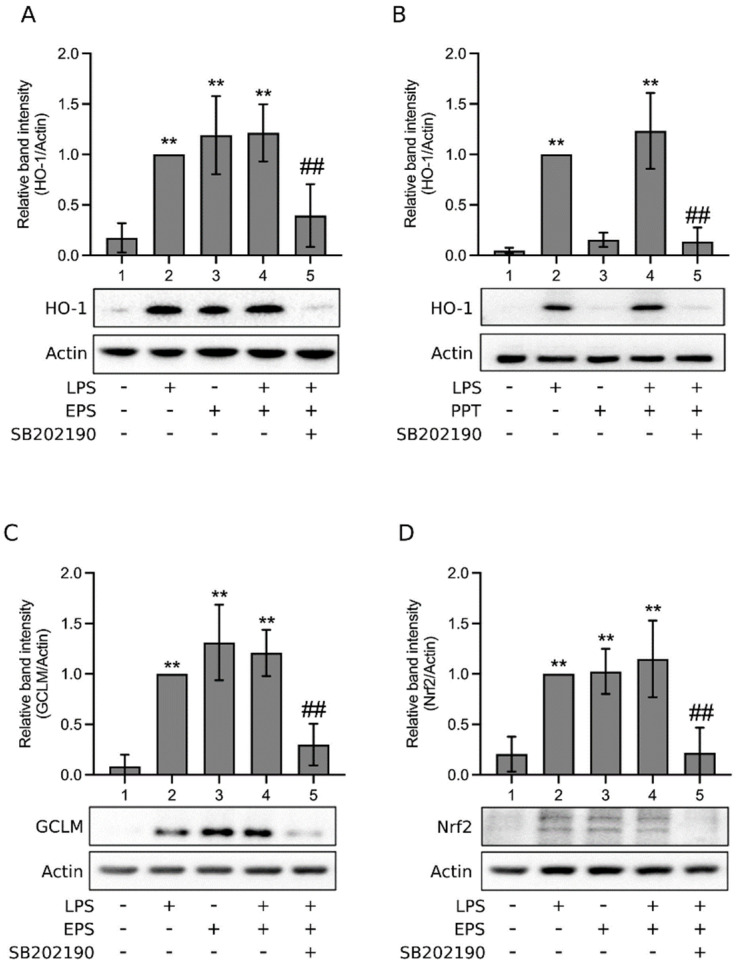
EPS activated the nuclear factor erythroid 2-related factor 2 (Nrf2)/heme oxygenase-1 (HO-1) pathway. (**A**,**C**,**D**) THP-1 cells were treated with vehicle (Lane 1), 1 μg/mL LPS (Lane 2), 100 μg/mL EPS (Lane 3), LPS and EPS (Lane 4), or LPS, EPS, and 20 μM SB202190 (Lane 5) for 24 h. (**B**) the same experiment of (**A**) was performed, but EPS was replaced by PPT. The expression levels of HO-1 (**A**,**B**), the glutamate-cysteine ligase modifier subunit (GCLM; **C**), and Nrf2 (**D**) in the cells were assayed using Western blotting. Relative band intensity vs. Lane 2 was determined. Data are presented as the mean ± standard deviation of four independent experiments. ** *p* < 0.005 vs. Lane 1; ## *p* < 0.005 vs. Lane 4.

**Figure 7 ijms-23-10237-f007:**
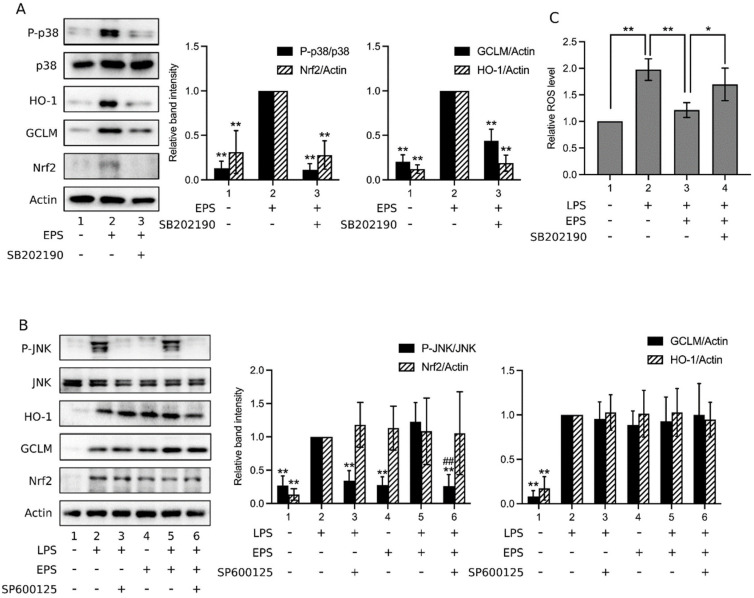
P38 inhibitor but not JNK inhibitor suppressed the activation of Nrf2/HO-1 pathway. (**A**,**B**) THP-1 cells were treated for 24 h with 1 μg/mL LPS alone, 100 μg/mL EPS alone, or the combinations of LPS, EPS, 20 μM SB202190, or 20 μM SP600125 (a JNK inhibitor) as indicated underneath the blots. The levels of phosphorylated p38, total p38, HO-1, GCLM, Nrf2, actin, phosphorylated JNK, and total JNK were assayed using Western blotting. Relative band intensity vs. Lane 2 was determined. Data are presented as the mean ± standard deviation of four independent experiments. * *p* < 0.05 and ** *p* < 0.005 vs. Lane 2; ## *p* < 0.005 vs. Lane 5 (**B**). (**C**) THP-1 cells were treated with vehicle (Group 1), 1 μg/mL LPS (Group 2), LPS and 100 μg/mL EPS (Group 3), or LPS, EPS, and 20 μM SB202190 (Group 4) and subjected to intracellular reactive oxygen species (ROS) assays. Relative ROS level vs. Group 1 was determined. Data are presented as the mean ± standard deviation of three experiments. * *p* < 0.05 and ** *p* < 0.005 between the indicated groups.

**Figure 8 ijms-23-10237-f008:**
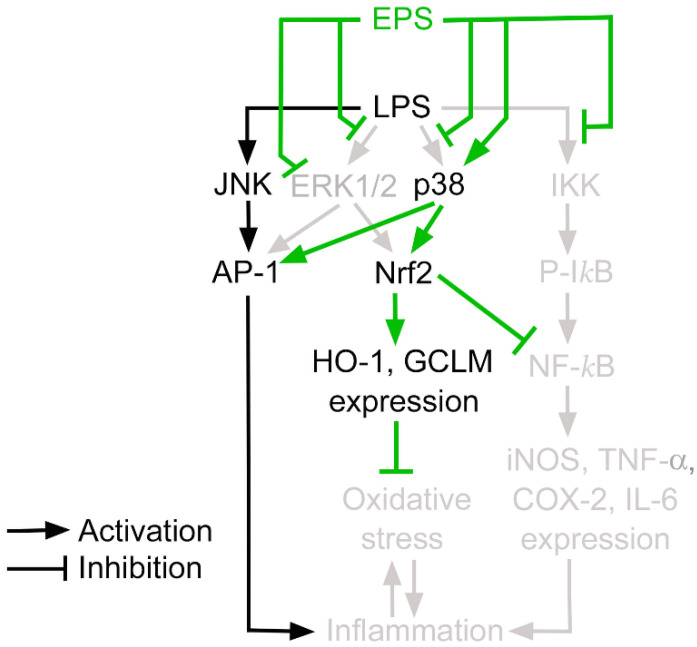
Schematic diagram displays the proposed mechanisms underlying the anti-inflammatory effect of EPS. The parts inhibited by EPS in the pathways are indicated by using light gray color.

## Data Availability

All data are reported in the manuscript.

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
