# Peer review of "Exopolysaccharides of Bacillus amyloliquefaciens Amy-1 Mitigate Inflammation by Inhibiting ERK1/2 and NF-κB Pathways and Activating p38/Nrf2 Pathway"

_ijms, 2022, doi:10.3390/ijms231810237_

Round 1

Reviewer 1 Report

The title is too long. The authors should change it and significantly shorten it. It looks very bad and needs to be corrected. The authors did not make any changes to the manuscript (changes should be added in a different color). I do not accept this manuscript for publication as it stands.

Author Response

Thank You for reviewing our manuscript. Please see the attached file for our point-to-point responses.

Reviewer 2 Report

In this research article the authors investigated the anti-inflammatory activity of EPS derived from Bacillus amyloliquefaciens, in vitro and in vivo. To this aim, the authors performed a thorough investigation of the pathways involved in this property, reaching the conclusion that EPS acts by inhibiting the NF-kB and the ERK1/2 pathways, while activating p38 signaling. These events also lead to the downregulation of inflammation-related oxidative stress, in activated THP-1 cells.

Points in the manuscript that require attention:

Lines 350-351: why did PPT inhibit the growth of the cells? Could the authors elaborate more on this, in the text?

A general comment on the Discussion section is that the authors could condense the results of the study and present them more concisely and include more literature references to critically assess their findings.

Lines 427-428: Please confirm that no antibiotics were added to the complete cell medium.

Line 432: it is important to define the “vehicle” in each experimental procedure. Is it different from PPT? Please make appropriate corrections in the text and in the legends of the figures.

Line 472: how many cells were seeded in the plates? What was the incubation period used to test for the expression of each different protein? Please add this information in the materials and methods section.

Line 496: Figure 2 does not include the results of treatments with LPS+SB202190 or EPS+SB202190. Rephrase this passage. It would be useful for the authors to include LPS+SB202190 in the figure for reference. Similarly, this treatment could be added in Figs 6C, D.

Figure 3: Please consider performing a statistical analysis of the difference between groups of EPS treatments – to support the dose-dependent reduction of ear edema (Lines 149-150).

Author Response

(The authors gave the same response as above.)

Reviewer 3 Report

The manuscript by Wei-Wen Sung et al. investigated the anti-inflammatory activity of exopolysaccharides from Bacillus amyloliquefaciens. The study is of interest to some of the readers and I have the following questions and suggestions:

1, The chemical structure of the exopolysaccharides must be provided and defined. The linkage mode, monosacchride composition, Mw etc. 

2, I have not seen the animal test results. The authors claimed that they have used eight-week-old male ICR mice for the animal experiments but why there are not results? This must be explained. 

3, The English of the whole manuscript must be further improved. The authors must carefully revise the whole manuscript. 

4, The strain name of Bacillus amyloliquefaciens must be provided. 

5, Is it possible for the EPS to bind to the TLRs of THP-1 cells? What do the authors think about this? What are the target proteins of EPS? These questions must be answered and discussed.

Author Response

(The authors gave the same response as above.)

Round 2

Reviewer 1 Report

I dont accept. 

Reviewer 3 Report

The authors have revised the manuscript accordingly. I suggest to accept it.